# Determination of the Relationships between the Chemical Structure and Antimicrobial Activity of a GAPDH-Related Fish Antimicrobial Peptide and Analogs Thereof

**DOI:** 10.3390/antibiotics11030297

**Published:** 2022-02-23

**Authors:** Samuel Cashman-Kadri, Patrick Lagüe, Ismail Fliss, Lucie Beaulieu

**Affiliations:** 1Institute of Nutrition and Functional Foods (INAF), Université Laval, Québec, QC G1V 0A6, Canada; samuel.cashman-kadri.1@ulaval.ca (S.C.-K.); ismail.fliss@fsaa.ulaval.ca (I.F.); 2Department of Food Science, Faculty of Agricultural and Food Sciences, Université Laval, Québec, QC G1V 0A6, Canada; 3Québec-Océan, Université Laval, Québec, QC G1V 0A6, Canada; 4Department of Biochemistry, Microbiology and Bioinformatics, Université Laval, Québec, QC G1V 0A6, Canada; patrick.lague@bcm.ulaval.ca; 5Institute for Integrative Systems Biology, Department of Biochemistry, Microbiology and Bio-Informatics, Pavillon, Alexandre-Vachon, Université Laval, 1045 Avenue de la Medecine, Québec, QC G1V 0A6, Canada; 6The Quebec Network for Research on Protein Function, Engineering, and Applications (PROTEO), 1045 Avenue de la Medecine, Québec, QC G1V 0A6, Canada

**Keywords:** antimicrobial peptides, GAPDH, structure–activity relationships, membrane permeabilization, SJGAP

## Abstract

The structure–activity relationships and mode of action of synthesized glyceraldehyde-3-phosphate dehydrogenase (GAPDH)-related antimicrobial peptides were investigated. Including the native skipjack tuna GAPDH-related peptide (SJGAP) of 32 amino acid residues (model for the study), 8 different peptide analogs were designed and synthesized to study the impact of net charge, hydrophobicity, amphipathicity, and secondary structure on both antibacterial and antifungal activities. A net positive charge increase, by the substitution of anionic residues or C-terminal amidation, improved the antimicrobial activity of the SJGAP analogs (minimal inhibitory concentrations of 16–64 μg/mL), whereas the alpha helix content, as determined by circular dichroism, did not have a very definite impact. The hydrophobicity of the peptides was also found to be important, especially for the improvement of antifungal activity. Membrane permeabilization assays showed that the active peptides induced significant cytoplasmic membrane permeabilization in the bacteria and yeast tested, but that this permeabilization did not cause leakage of 260 nm-absorbing intracellular material. This points to a mixed mode of action involving both membrane pore formation and targeting of intracellular components. This study is the first to highlight the links between the physicochemical properties, secondary structure, antimicrobial activity, and mechanism of action of antimicrobial peptides from scombrids or homologous to GAPDH.

## 1. Introduction

It is now well-documented that the massive use of traditional antibiotics for the treatment of diseases caused by different pathogenic microorganisms promotes the development of multi-drug resistant microorganisms that are capable of resisting several different antibiotics. In fact, the World Health Organization now considers antibiotic resistance one of the most serious threats to global health, food security, and development [1]. In this regard, antimicrobial peptides (AMPs), which can be defined as small peptide molecules, generally composed of 7 to 100 amino acids that show inhibitory activity against microorganisms, represent a very promising avenue in the search for alternative molecules to conventional antibiotics. AMPs are ubiquitous in the living world and an important part of the innate defense system of most organisms against microbial pathogens. Numerous studies conducted on these AMPs have shown that they most often act by destabilizing microbial membranes, notably by permeabilizing them, as evidenced by some reviews and articles written specifically on this subject [2,3,4]. Moreover, these mechanisms are generally not dependent on specific receptors, thus reducing the risks of resistance development and increasing the therapeutic potential of these molecules [5,6,7,8,9]. This certainly explains why AMPs have been the subject of so much research in recent years; between 2012 and 2017, an average of 12,000 scientific papers have been published annually, in relation to antimicrobial AMPs [6].

In addition to these potential therapeutic applications, AMPs have also been studied for their use in food preservation [10,11,12,13,14]. Indeed, they could promote food safety by combating the development of pathogens in food, which are responsible for several million foodborne illnesses each year, resulting in enormous human and monetary costs [15]. Moreover, AMPs have the ability to inhibit the growth of the microorganisms responsible for food spoilage and could potentially be used to improve food shelf life [16,17,18,19]. Nisin is currently the only antimicrobial peptide approved by Health Canada and the FDA as a food preservative, but pediocin has also been used commercially and is in the process of legal approval [20,21].

Fish are a very prolific source of AMPs, as evidenced by several literature reviews that focus specifically on antimicrobial peptides of this origin [22,23,24]. Indeed, fish generally possess a very strong immune system that allows them to develop in an environment where exposure to many pathogens is very high. Thus, fish can continuously fight these pathogens by secreting a wide variety of AMPs, which play a significant role in the innate defense of fish against bacteria, fungi, viruses, and other pathogens. These AMPs are often produced in the components of fish most likely to be exposed to pathogens, such as mucus from the skin, circulatory system, and saliva [24]. It is also interesting to note that some families of AMPs are found only in fish, such as piscidins, located in the gills of certain species [22]. In addition, some studies have been able to identify the specific peptides involved in the antimicrobial activity of hydrolysates from different fish, such as the Atlantic mackerel [25,26], filefish [27], anchovy [28], and barbel [29]. It has also been shown that certain bacteria, present in the digestive system of fish, produce antimicrobial compounds, such as formicin, which is a lantibiotic-type bacteriocin [30].

In order to be able to identify and develop AMPs that could be used as therapeutic agents or food preservatives, research of a more fundamental nature is necessary, prior to work that more directly concerns their concrete applications. Certainly, one of the most important aspects of this more fundamental aspect is the study of the structure–activity relationships (SAR) of these peptides, i.e., the study of the links between their chemical structure (primary and secondary structures), physicochemical properties (size, net charge, hydrophobicity, amphipathicity, etc.), and biological activity. These studies make it possible, for example, to identify the key regions or amino acids and structural properties required for the antimicrobial activity of peptides. This type of work is also very closely related to the study of the mechanisms of action of these peptides, particularly with regard to their interactions with cell membranes. These structural and mechanistic studies are, therefore, essential for the optimization of AMPs, which is required for the development of antimicrobial agents of pharmaceutical or food interests [12,31,32]. In fact, several SAR studies have been carried out on the AMPs derived from fish. For example, studies on pleurocidin show that its antimicrobial activity is linked to its hydrophobic N-terminal residues and helical structure, which is involved in membrane interactions [33,34,35]. Moreover, the antimicrobial activity of piscidin-1 has been linked to its net charge, its helical structure, and specific phenylalanine residues [36,37].

Recently, a low molecular weight peptide, active against the pathogenic bacterium *Listeria monocytogenes,* the Atlantic Mackerel glyceraldehyde-3-phosphate dehydrogenase (GAPDH)-related derived peptide (AMGAP), has been identified and characterized from an enzymatic hydrolysate of Atlantic mackerel [38]. This antilisterial peptide was found to have sequence homology with two AMPs derived from marine fish of the same family (*Scombridae*), identified as the Yellowfin Tuna GAPDH-related antimicrobial peptide (YFGAP) [39] and the Skipjack Tuna GAPDH-related antimicrobial peptide (SJGAP) [40], which have a much broader spectrum of activity. These three peptides from *Scombridae* are also related to the GAPDH enzyme, which is widely distributed in the living world and directly involved in the metabolic pathway of glycolysis. However, not only have no SAR studies been performed on these three peptides, but AMPs homologous to GAPDH have received little attention in the scientific literature. Among the few studies carried out on AMPs related to this enzyme, antifungal activities have been highlighted, showing the potential role of GAPDH in the defense system of various organisms [41,42].

The aim of this study was to highlight links between the chemical structure of the GAPDH-related peptides derived from scombrids and their antibacterial and antifungal activities. The main objectives were to (i) synthesize SJGAP and relevant analogs thereof, (ii) assess their antimicrobial activity, (iii) characterize their physicochemical properties and secondary structure, and (iv) determine their membrane permeabilization ability, so that (v) precise links between their chemical structure, antimicrobial activity, and mechanism of action could be established.

## 2. Results

### 2.1. Physicochemical Properties

Peptides’ molar weights, net charges, and GRAVY indexes are shown in Table 1. All analogs are positively charged, except analog 4 (C-terminal portion of SJGAP), which has a zero net charge. Analogs 5 and 6 have a +2 and a +4 charge, compared to the native SJGAG, respectively, due to anionic glutamic acid (E_26_) and aspartic acid (D_32_) residues that have been substituted for neutral alanine (A) (analog 5) and cationic lysine (K) residues (analog 6). All analogs, except analog 4, have positive GRAVY values, which indicates that these peptides are hydrophobic. The negative GRAVY value of analog 4 means that it is more hydrophilic. Analog 5 has the highest GRAVY value, which is due to the substitution of the hydrophilic E_26_ and D_32_ residues with hydrophobic A residues. Analog 7 has the second-highest GRAVY value amongst the full-length (32 amino acid residues) analogs because of the three A_18_, A_19_ and A_29_ that have been substituted with more hydrophobic leucine (L) residues.

### 2.2. Antimicrobial Activity

Minimal inhibitory concentrations (MICs) and bactericidal concentrations (MBCs) of peptide analogs against bacterial strains are shown in Table 2, while their MICs and minimal fungicidal concentrations (MFCs) against fungal strains are shown in Table 3. Firstly, native SJGAP (analog 1) showed no antibacterial activity against the seven tested strains. However, it showed low activity (MIC = 128 μg/mL) against *Paecilomyces* sp. (filamentous fungus) and *Z. rouxii* (yeast). SJGAP’s N-terminal segment (analog 2) and middle segment (analog 3) showed no antimicrobial activity, but the C-terminal segment (analog 4), while showing no antibacterial activity, showed a very limited antifungal activity, with a MIC value of 64 μg/mL, against the *Z. rouxii* yeast strain. Analogs 5 and 6 showed the strongest antimicrobial activity. Indeed, they both showed strong inhibitory activity against all tested bacterial strains, with MIC values ranging from 16 to 64 μg/mL. Both of these analogs also showed bactericidal activity against most Gram-negative strains (MBCs ranging from 32 to 128 μg/mL), but showed to be bacteriostatic against Gram-positive strains. These two analogs also showed strong antifungal activity against *Paecilomyces* sp., *Z. rouxii*, and *R. mucilaginosa* (MICs of 32–64 μg/mL), including fungicidal effect on the latter strain (MFC = 64 μg/mL). Analog 7, while showing no antibacterial activity, was very active against the filamentous fungus *Paecilomyces* sp. (MIC = 16 μg/mL) and the yeasts *Z. rouxii* (MIC = 64 μg/mL) and *R. mucilaginosa* (MIC = 32 μg/mL). It also showed fungicidal activity against the latter yeast (MFC = 64 μg/mL). Analog 8 showed inhibitory activity against all tested Gram-negative bacterial strains, except for *A. salmonicida* 69 R5, exhibiting strong bactericidal activity against the two *E. coli* strains (MBC = 32–64 μg/mL) and bacteriostatic activity against *P. aeruginosa* (MIC = 64 μg/mL) and *A. salmonicida* 69 R3 (MIC = 128 μg/mL). Analog 8 showed no activity against Gram-positive strains, but showed to be inhibitory toward the *Paecilomyces* sp. and *Z. rouxi* fungal strains (MIC = 64 μg/mL). Chloramphenicol was used as a reference antibiotic and showed a 4 μg/mL MIC against the *E. coli* ATCC 25922 reference strain.

### 2.3. Secondary Structure

#### 2.3.1. Circular Dichroism (CD) Measurements

CD spectroscopy studies in the far-UV region (190–260 nm) can provide useful insights on the secondary structure of peptides and proteins. The CONTIN algorithm is known to provide accurate CD spectrum analysis, especially in regard to the alpha helix and beta strands contents [43]. CD spectra of SJGAP and analogs 2–8 (Figure 1,) and their secondary structures relative contents, obtained with the CONTIN algorithm (Table 4), show that in aqueous environment (phosphate buffer (PB), 10 mM), all peptides display very low helicity and are constituted mostly by beta strands and random coils (unordered structure). Indeed, beta strands are typified by a maximum around 195 nm and minimum around 218 nm [44], while unordered structures are typically characterized by a broad spectrum and negative shoulder, around 200 nm [45,46]. The presence of 25% trifluoroethanol (TFE) increases the beta strand content for most analogs (except analogs 2 and 4), but does not increase their alpha helix content. When TFE concentration is increased to 50% and 75%, the alpha helix content of analogs 3, 6, 7, and 8 starts to increase, at the expense of the beta strands content, as it can be seen by the two characteristic minima at 208 and 222 nm and the maximum at 193 nm [44]. Random coil content is not affected by this change in solvent.

#### 2.3.2. Secondary Structure Predictions

PSIPRED uses position-specific amino acid sequence alignments (BLAST), in combination with neural networks, to produce accurate secondary structure predictions of peptides and proteins. Results obtained for analogs 1, 5, 6, and 7 are presented in Figure 2. Predicted structures are very alike for all peptides, and only some minor variations (±one residue) in the length of the N-terminal beta strand, central alpha helix, and C-terminal beta strand can be observed. PSIPRED’s confidence levels for the predicted structures are globally very high, except for residues H_21_ and N_31_, which are at the C-terminal end of the central alpha helix of the peptides and C-terminal end of the C-terminal beta strand, respectively.

### 2.4. Membrane Permeabilization

#### 2.4.1. SYTO 9/Propidium Iodide (PI) Staining

Results obtained from the Invitrogen LIVE/DEAD^®^ BacLight Bacterial Viability kit are presented in Figure 3 as SYTO 9/PI fluorescence ratios for *E. coli* ATCC 11229, *M. luteus* LMA-272, and *R. mucilaginosa* 27173. For the Gram-negative *E. coli* ATCC 11229 (Figure 3a), all five tested peptides, at both concentrations, caused membrane permeabilization, as shown by the significantly smaller SYTO 9/PI ratios than for the negative control (NaCl 0.85% solution). All peptides used at MIC caused less permeabilization than when used at MBC or 2x MIC, but not at a statistically significant level. Analogs 5, 6, and 7 caused membrane permeabilization equivalent to that of the positive control (ethanol 50%). For the Gram-positive *M. luteus* LMA-272 (Figure 3b), analogs 1 and 8 did not cause any significant membrane permeabilization, while analogs 5, 6, and 7 did, as they are associated with significantly smaller SYTO 9/PI ratios than for the negative control. As for *E. coli*, using the MIC or 2x MIC caused equivalent membrane permeabilization for all of the peptides. The permeabilization caused by analogs 5 and 7 (at both concentrations) and analog 6 (MIC) is equivalent to that of the positive control, but the latter used at 2x MIC produced a significantly higher SYTO 9/PI ratio than the positive control, thus indicating less membrane permeabilization. Results obtained for the yeast *R. mucilaginosa* (Figure 3c) indicate that all peptides, except for analogs 1 and 8, used at their MIC, caused significant membrane permeabilization.

#### 2.4.2. 260 nm-Absorbing Intracellular Material Leakage

Ratios of 260 nm optical density values, after the 120 min incubation period, were compared to the values measured at T0 (data not shown). For all tested peptide analogs (both concentrations) and each of the three tested microorganisms, no significant 260 nm-absorbing material leakage outside of the cells has been detected, as the 120-minute incubation period did not significantly increase the OD value measured at 260 nm, compared to that obtained with the negative control (NaCl 0.85% solution). 

## 3. Discussion

AMPs have been identified in multiple fish species, as highlighted by several reviews on this topic [22,23,47]. Very recently, the antibacterial activity of a small peptide from a hydrolysate of Atlantic mackerel, named AMGAP, has been demonstrated [38]. This peptide is homologous to two other AMPs that were studied by Seo et al., namely YFGAP [39] and SJGAP [40]. These three scombrids peptides are homologous to the N-terminal segment of the GAPDH, which is known to be a precursor of other AMPs [41,42,48,49]. However, to the best of our knowledge, no SAR studies have been performed on these peptides from scombrids or derived from GAPDH. The results obtained in this study allow us to highlight several links between the physicochemical properties of the peptide analogs, as well as their antimicrobial activity and secondary structure. Some information, related to the mechanism of action of these peptides, can also be derived from the two membrane permeabilization experiments performed in this study.

### 3.1. Antimicrobial Activity and Peptides’ Sequences

Since SJGAP’s C-terminal portion is strongly homologous to the AMGAP sequence, and that it has shown strong antimicrobial activity against Gram-negative bacteria, Gram-positive bacteria and the yeast *Candida albicans* [40], it has been selected as the model peptide for this study. The first thing to note is that, contrary to Seo et al. [40], we could not demonstrate antibacterial activity associated with native SJGAP, even though we tested three of the same bacterial species (*E. coli*, *P. aeruginosa*, and *M. luteus*). The strains are not the same, but the assumption of strain-specific inhibitory activity of SJGAP is unlikely given that AMPs generally have mechanisms of action that are not directed at a specific molecular target [2]. In fact, it is widely accepted that most AMPs act on the cytoplasmic membrane of targeted microorganisms, causing pore formation or complete membrane destabilization, leading to loss of intracellular material and cell dysfunction [4]. This membrane destabilization is generally not linked to specific structures that might be strain-specific and depends, rather, on the chemical interactions between peptides and major microbial membrane components (notably phospholipids), which is in contrast to the mechanism of some particular AMPs that interact with specific molecular targets (such as lipid II for nisin and mutacin), hence explaining the narrower activity spectrum of these two bacteriocins [50].

A more likely hypothesis is related to the fact that synthetic SJGAP were used, rather than extracted and purified (HPLC) from skipjack tuna skin, to perform the antimicrobial tests. Indeed, despite efficient purification, it is possible that other compounds eluted simultaneously with SJGAP have a synergistic effect on its antimicrobial activity [51]. Such a synergistic effect has also been proposed to explain the higher activity of purified AMGAP, compared to the synthesized one [38]. However, our results show that the native SJGAP (analog 1) is active against a yeast (*Z. rouxii*) and filamentous fungus (*Paecilomyces* sp.), confirming the antifungal activity detected by Seo et al. [40].

The microbial strains used in this study were selected to thoroughly test the spectrum of activity of the peptides and their potential for therapeutic and food applications. *E. coli* ATCC 25922 is a reference strain widely used in the literature and, therefore, allows us to rigorously compare the activity of SJGAP and its analogs with other antimicrobial peptides and compounds. *P. aeruginosa* ATCC 27853 and *L. ivanovii* ATCC 19119 are two bacterial strains known to have pathogenic potential, as attested by the biosafety level 2, assigned by ATCC. Then, two strains of *Aeromonas salmonicida* were selected, as this bacterial species represents a major threat to salmonid fish farming and stimulates the search for alternatives to the use of antibiotics, in this context [52,53]. Finally, *M. luteus* LMA-272, as well as the three yeasts and three molds tested, are all strains that have been isolated from food products, thus allowing us to study the food application potential of the peptides tested.

Seven peptide analogs have been designed and synthesized, based on the 32-amino acid residues sequence of the native SJGAP. Peptide analogs 2, 3, and 4 were synthesized to assess whether SJGAP’s activity was directly linked to a particular region in the peptide chain. Indeed, while the AMGAP sequence is homologous to the C-terminal end of the SJGAP, it has been shown in another study that a 12-residue, C-terminal fragment of the fish antimicrobial peptide pleurocidin displayed an antimicrobial activity, comparable that of the native pleurocidin [54]. This indicates that some fragments may possess activity equivalent to that of the native peptide from which they are derived. Results (presented in Table 2 and Table 3) show that none of these three fragments displayed antimicrobial activity, which is not surprising, considering the very low antimicrobial activity that could be found for the native SJGAP. Surprisingly, however, the C-terminal fragment of SJGAP (analog 4) showed evident inhibitory activity (MIC = 64 μg/mL) against the yeast *Z. rouxii*, even more so than the native SJGAP (MIC = 128 μg/mL). According to this result, it would certainly be interesting to see if the AMGAP, highly homologous to this fragment, has the same activity against this yeast. Given the lack of conclusive antimicrobial activity of these three SJGAP fragments, they were not included in the membrane permeabilization experiments.

Peptide analogs 5 and 6 were designed to evaluate both the impact of negatively charged residues and the increase of the net charge on SJGAP’s activity. As a matter of fact, the importance of net charge for the antimicrobial activity of numerous peptides has been highlighted, whether they come from fish [9,32,36,37] or other sources [55,56]. It can also be noted that anionic residues, such as aspartic acid (D) and glutamic acid (E), are rarely present in the sequences of AMPs [57].

Analog 7 was conceptualized, in order to investigate the impact of increasing the hydrophobicity of the hydrophobic face of the alpha helix of the SJGAP structure, as obtained by homology modeling and represented by a Schiffer-Edmundson wheel projection [40]. This also has the direct consequence of increasing the amphipathicity of the analog. The hydrophobicity of the C-terminal end was also slightly increased with the substitution of the A_29_ residue with a leucine residue. In fact, hydrophobicity and amphipathicity have been extensively linked with the inhibitory activity of several AMPs [35,58,59,60,61]. 

The C-terminal amidated SJGAP (analog 8) has been synthesized because C-terminal amidation is a very common post-translational modification of AMPs, which can have a major impact on antimicrobial activity [62]. In fact, C-terminal amidation raises both the net charge of the peptide (+1) and its hydrophobicity, because it suppresses the negative charge bore by the carboxyl group on the free C-terminal end [63]. Besides, it has been shown that a C-terminal amidation can have a stabilizing effect on the structural conformation of peptides, which has already been correlated with their antimicrobial activity [64].

The MIC values presented in Table 2 and Table 3 show that some of these analogs have considerable antimicrobial activity against several bacterial and fungal strains. These values indicate moderate antimicrobial activity [65], but the MIC value of 16 μg/mL, shown by analogs 5 and 6 against two bacterial strains and analog 7 against the fungal strain *Paecilomyces* sp., is close to the very interesting activity threshold set at 10 μg/mL by Ríos and Recio (2005) [66]. From the above results, it can be deduced that the net charge of the analogs is strongly correlated with their antibacterial activity. The two most cationic analogs, which are analogs 5 and 6 (net charges of +6 and +8, respectively), proved to be the most active against all the tested bacterial strains. Analog 8, having a charge of +1, compared to the native SJGAP, also showed inhibitory activity against four of the five Gram-negative bacterial strains, but was inactive against the Gram-positive strains. For example, the +1 net charge, caused by the C-terminal amidation, also increased the antibacterial activity of pteroicidin-α, another AMP from fish [64]. However, analog 7, with a net charge (+4) equal to that of native SJGAP, did not show antibacterial activity. 

Several points can be made here. Bacterial membranes are well-known to contain negatively charged phospholipids, and the outer membrane of Gram-negative bacteria contains lipopolysaccharides (LPS), which are also anionic in nature. These strongly electronegative membranes, thus, allow the initial electrostatic interactions with cationic peptides, explaining why it has been observed that an increase in the cationicity of AMPs is often correlated with an increase in their inhibitory activity [4,9]. This is true regardless of whether the peptides have the microbial membrane as a direct target or intracellular targets, in that these peptide interactions with the membrane are prerequisites for all these mechanisms [57]. It can also be noted that the distribution of charges on the peptide sequence can have an impact on its activity [3,4]. Thus, it is possible that the reduction of the anionic character of the C-terminal portion of SJGAP, by the substitution of anionic residues E_26_ and D_32_ with neutral alanine residues (analog 5), cationic lysine residues (analog 6), or by C-terminal amidation (analog 8), is important if this portion is critical in the initial attachment of the peptide to the membrane, which will be studied in subsequent work. Furthermore, it is known that the cationic amino acids lysine and arginine are very common in the sequences of AMPs, as they are essential for electrostatic interactions with the anionic components of microbial membranes [57].

Regarding the superior inhibitory activity of the peptides against Gram-negative bacteria, it can first be evoked the presence of LPS on the outer membrane, which can favor the first interactions with the cationic peptides. Gram-positive bacteria, on the other hand, have a cell wall consisting of a thick layer of peptidoglycan, in which lipoteichoic acids (LTAs) are present, which are also anionic polymers. The interactions between peptidoglycan and AMPs are still relatively unclear, but it seems that it does not have a negative impact on the penetration of AMPs and their interactions with the phospholipids of the cytoplasmic membrane, given its relatively high porosity. However, LTAs could very well retain cationic peptides and affect their interactions with the cytoplasmic membrane, thus offering a possible explanation for their lower activity against Gram-positive bacteria [67,68].

The results obtained with the two *A. salmonicida* strains also tend to support this link between the net charge of the analogs, their interactions with membranes, and their inhibitory activity. Indeed, the only difference between the 69 R3 and 69 R5 strains is the presence of the A-proteins that form the A-layer on some strains of this bacterium [69,70]. These A proteins are not only quite hydrophobic, but they also have an acidic isoelectric point [69], which, therefore, translates into their anionic nature. It is, thus, possible that this more anionic surface of the 69 R3 strain of bacteria could explain the greater activity of the cationic SJGAP analogs, compared to the 69 R5 strain of bacteria, which lack this protein layer.

The impact of the positive net charge seems somewhat less direct, with respect to the antifungal activity of SJGAP analogs. Firstly, it is interesting to note that native SJGAP is weakly active against two fungal strains, including a mold (*Paecilomyces* sp.) and yeast (*Z. rouxii*), while this activity is slightly enhanced by the C-terminal amidation (analog 8), resulting in a two-fold lower MIC value for the same strains. Analogs 5, 6, and 7 showed the strongest antifungal activity, not only noticeable by the lower MIC values for the same two strains, but also by their inhibitory activity against the yeast *R. mucilaginosa*. The high antifungal activity of analog 7, with a net charge equal to that of native SJGAP, thus shows that the cationicity of the analogs is not the only factor to be considered in explaining their antifungal activity.

In this respect, the hydrophobicity of the peptides also seems to play an important role. Similarly to what was observed in the present study, such an increase in hydrophobicity by the substitution of residues on the hydrophobic face of the alpha helix of AMPs with more hydrophobic residues has already been shown to be favorable for their antifungal activity [56]. The differences in composition between bacterial and fungal membranes may offer an interesting avenue for explaining this result. Indeed, even if fungal membranes contain a non-negligible proportion of anionic phospholipids (notably phosphatidylserine and phosphatidylinositol), the surface of eukaryotic cells is generally much less negatively charged than that of prokaryotes, which contain a greater proportion of anionic lipids, such as phosphatidylglycerol and cardiolipin [4,71]. In fact, the major phospholipids in fungal membranes, phosphatidylcholine, and phosphatidylethanolamine are zwitterionic in nature [61]. It is also relevant to mention the presence of ergosterol in the fungal membrane, which is a hydrophobic molecule absent in bacterial membranes. It is, therefore, likely that the less anionic nature of the fungal membrane compared to bacterial membranes may favor hydrophobic interactions with analog 7, which could explain why it has inhibitory activity against fungi but not bacteria.

### 3.2. Secondary Structure

The antimicrobial activity of many peptides is clearly linked to their secondary structure. In fact, the spatial conformation of peptides has a direct impact on their amphipathicity: certain secondary structure patterns, notably alpha helices and beta strands, promote the amphipathic character of peptides [4]. These structural motifs are the key to the activity of most AMPs, despite their great structural diversity [5]. However, many AMPs are unstructured in aqueous conditions or before contact with microbial membranes [72,73]. In this respect, alcohols, in particular TFE, are often used to simulate the membrane environment and well-known to induce alpha helix formation, which is why this solvent is frequently used in SAR studies of AMPs [32,58].

In this study, the secondary structure of SJGAP, and analogs thereof, was investigated by CD with different concentrations of TFE and by a bioinformatics prediction method. As expected, in aqueous media (PB), as well as in the presence of a low concentration of TFE (25%), all peptides showed low helicity, but the structures of analogs 6 and 8 are strongly affected by the increase in TFE concentration, which causes the formation of the central alpha helix, as predicted by PSIPRED. Indeed, it can be seen that the central alpha helix of analogs 1, 5, 6, and 7, as predicted by PSIPRED (see Figure 2), is formed by 12 residues (11 for peptide 6), which corresponds quite precisely to the alpha helix contents of peptide 6 (41%) and peptide 8 (36%), as determined by CD in 75% TFE. 

However, the same helix predicted for analogs 1, 5, and 7 was not observed with CD. This is not necessarily surprising, as PSIPRED is a multiple sequence alignment (MSA)-based method [74], which, in the present situation, used the sequences of several GAPDHs, given the high homology between SJGAP and this enzyme. This explains why the sequences of the four analogs submitted to PSIPRED have almost exactly the same predicted secondary structures, as the homology modeling was based on the same sequence alignments of different GAPDHs. This shows how useful it is to combine different secondary structure analysis methods for the analysis of peptides that are produced by the cleavage of a protein, whose secondary structure is already well-known.

In contrast to what has been described elsewhere [9], the present study did not detect a clear correlation between the helical structure of the peptides and their antimicrobial activity. Indeed, analog 5, which is very weakly helical, according to the CD experiments, showed antibacterial and antifungal activities similar to those of analog 6 and superior to those of analog 8, which has a much more helical structure. Similarly, analog 7 showed high antifungal activity, despite its low helicity. It should be noted, however, that the antifungal activity of peptides has been previously observed to not be as dependent on their helicity as for their antibacterial activity [75]. Moreover, SJGAP is not a predominant alpha-helical peptide, as are other AMPs from fish, such as piscidin-1 [76], pleurocidin [77], or myxinidin [32]. It is, therefore, very possible that both beta strands of SJGAP and its analogs are strongly involved in interactions with microbial membranes, thus explaining why the central alpha helix is not essential for their activity.

### 3.3. Membrane Permeabilization

To obtain information on the mode of action of SJGAP and its analogs, two experiments to assess their membrane permeabilization capacity were conducted. The LIVE/DEAD BacLight^TM^ Bacterial Viability kit consists of two fluorescent dyes, namely SYTO 9 and PI, and is commonly used to evaluate AMPs’ ability to disrupt microbial membranes [78]. These two dyes, when bound to nucleic acids, emit a high fluorescence (green for SYTO 9 and red for PI), but do not have the same membrane permeability properties. Indeed, while SYTO 9 can penetrate intact membranes, as well as membranes with compromised integrity, PI cannot pass through intact membranes, but only through disrupted membranes. Furthermore, when both dyes cross the membrane, SYTO 9 is dislodged by PI, due to the higher affinity of the latter for nucleic acids. It is then possible to evaluate the membrane integrity of microorganisms by measuring the relative intensity of these two fluorescent dyes [79,80]. We also wanted to evaluate the degree of membrane permeabilization caused by the peptides by measuring whether they produce a loss of intracellular material absorbing at 260 nm, mainly DNA and RNA.

The fact that all tested peptides caused a significant level of membrane permeabilization on *E. coli*, as assessed by the BacLight test, but that native SJGAP and analog 7 did not show any inhibitory activity against this microorganism, requires some consideration. Likewise, analog 7 caused significant membrane permeabilization of *M. luteus* cells, without having any inhibitory activity. Firstly, it is worth noting that these results are somewhat contradictory to those obtained by Seo et al. [40], who, using a different methodology, concluded that SJGAP did not really induce membrane permeabilization in *E. coli*. Secondly, the fact that SJGAP and analog 7 showed membrane permeabilizing activity against one or both bacteria, without revealing quantifiable inhibitory activity in microplates is surprising, but some hypotheses can be put forward. It is generally accepted that PI-stained cells have their cytoplasmic membrane permeabilized and are, therefore, non-viable [81]. However, some cases in which bacteria stained red by PI without being non-viable have been described in the literature. Indeed, it has been reported that extracellular nucleic acids can induce PI fluorescence, thus causing an underestimation of the viability of the cells studied, but this was observed in the case of biofilms [82]. Moreover, cases of permeabilized membrane repair have been reported in both bacteria and yeast under stressful conditions, highlighting the possibility of PI staining, without signifying cell non-viability [83,84].

Another avenue of explanation concerns the nature of the permeabilization, induced by SJGAP and its analogs. Membrane permeabilization caused by AMPs is generally classified into three mechanisms of action: barrel-stave pores, toroidal pores (wormholes), and the carpet model. However, other alternatives are possible, in the case of non-lytic mechanisms of action, i.e., not directly based on membrane disruption. Indeed, some peptides can permeabilize the cytoplasmic membrane, without damaging, it by forming transient pores, which can then allow peptides to reach the cytoplasm and target intracellular components [57]. In fact, some peptides are even specifically dependent on pore disintegration to cross the membrane [3].

In this regard, it is relevant to mention that Branco et al. found that other AMPs (saccharomycin and synthetic analogs) homologous to GAPDH not only induce permeabilization of yeast cytoplasmic membranes, as evidenced by PI uptake, but that these peptides are also internalized in the cytoplasm of these yeasts [85]. These cells also showed specific markers related to apoptosis, supporting the hypothesis of intracellular mechanisms. However, the mechanism of action of the peptides, studied by Branco et al., should not be linked too directly with those of SJGAP and its analogs, because the latter are homologous to the N-terminal portion of GAPDH, whereas the former are homologous to the C-terminal segment of this enzyme [41]. Still, this clearly shows the possibility of membrane permeabilization that allows intracellular action of AMPs, rather than direct lethal membrane disruption. Thus, although analogs 1 and 7 can induce membrane permeabilization and possibly enter the cytoplasm of bacteria, they may not be able to interact with intracellular targets, as analogs 5, 6, and 8 do, which would explain their membrane permeabilizing activity, without showing inhibitory activity.

The latter hypothesis would also be consistent with the results obtained with leakage assays of intracellular absorbent material at 260 nm. Indeed, no significant leakage was detected for the three microorganisms, in contact with SJGAP and analogs thereof. This clearly shows that these peptides do not cause a complete disintegration of the membranes, thus showing that a carpet model mechanism of action is not possible. Such a result is, however, very compatible with the possibility of small temporary pores caused by the peptides. DNA and RNA polymers, which are large molecules, would not have the possibility to leave the cytoplasm of cells through these small, short-lived pores. These results also confirm the findings of Seo et al., who showed that SJGAP did not cause calcein to leak out of phospholipid liposomes [40]. The hypothesis of mixed mechanisms cannot be ruled out either: it is possible that the antimicrobial activity of SJGAP analogs is due to both membrane permeabilization (leakage of ions and other small molecules) and their action on intracellular targets. Such a mixed mode of action would also allow us to account for the bactericidal (against Gram-negatives) and fungicidal action of the peptides, as well as for their bacteriostatic action against Gram-positives, because it is known that membrane mechanisms normally induce cell death, while some intracellular targets, notably DNA and RNA, are, rather, associated with a bacteriostatic action [4,86].

However, it should also be noted that for the Gram-positive *M. luteus*, analogs 1 (native SJGAP) and 8, which caused no significant membrane permeabilization, also showed no inhibitory activity. Moreover, analogs 5 and 6, having shown strong inhibitory activity, also caused membrane permeabilization in this microorganism. The same expected relationship between membrane permeabilization and inhibitory activity can be observed for *R. mucilaginosa*, but to an even greater degree: for this yeast, analogs 1 and 8, which showed only weak inhibitory activity, induced membrane permeabilization when used at their MFC, but not at their MIC, while analogs 5, 6, and 7 caused both strong membrane permeabilization (MICs and MFCs) and growth inhibition in microplates. This positive correlation between the degree of membrane permeabilization and inhibitory activity highlights the fact that peptide insertion into membranes and pore formation, transient or not, are important for antimicrobial activity, whether due to direct disrupting action on the membrane or non-lytic mechanisms targeting intracellular structures [57].

## 4. Materials and Methods

### 4.1. Peptide Design and Synthesis

SJGAP [40] was chosen as the model peptide for this study. Seven analogous peptide sequences were designed, as shown in Table 5. Peptide analogs 2 to 8 were designed, in order to study SAR of the SJGAP. Peptides 2, 3, and 4 are N-terminal, middle, and C-terminal segments of the native SJGAP, respectively. Peptides 5 and 6 were designed by substituting the glutamic acid (E_26_) and aspartic acid (D_32_) residues with neutral alanine residues (analog 5) or positively charged lysine residues (analog 6). Peptide 7 was conceptualized by substituting alanine residues 18, 19, and 29 with leucine residues. Peptide 8 is the C-terminal amidated SJGAP.

Native SJGAP and analogs thereof were synthesized by the external company Bio Basic (Markham, Canada), with a >95% purity, as confirmed by HPLC/MS reports. Lyophilized peptide powders were kept at −20 °C until use.

### 4.2. Determination of Peptides Physicochemical Properties

Peptide molar weights, net charges, and GRAVY indexes were calculated using the ProtParam tool of Expasy bioinformatics resource portal (https://web.expasy.org/protparam/ (accessed on 30 January 2022)).

### 4.3. Antimicrobial Activity Assays

#### 4.3.1. Microbial Strains and Culture Conditions

All microbial strains were maintained in 25% (*v*/*v*) glycerol at −80 °C until use. *Escherichia coli* ATCC 25922, *Escherichia coli* ATCC 11229, *Pseudomonas aeruginosa* ATCC 27853, *Aeromonas salmonicida* subsp. *salmonicida* 69 R3, *Aeromonas salmonicida* subsp. *salmonicida* 69 R5 [53], *Micrococcus luteus* LMA-272, and *Listeria ivanovii* ATCC 19119 were used for antibacterial activity testing. Details on these strains used in this study are included in Table 6. All these strains were cultivated in tryptic soy broth (TSB) (BD, Sparks, NV, USA), except for *L. ivanovii*, which was grown in brain heart infusion (BHI) broth (BD, Sparks, USA). All bacterial strains were grown at 37 °C, except for the two *A. salmonicida* strains, which were incubated at 20 °C. Strains were sub-cultured at least twice, before being used for antimicrobial and membrane permeabilization assays. 

*Mucor racemosus* LMA-722, *Paecilomyces* sp. 5332-9a, and *Aspergillus niger* 3071-13 were used as filamentous fungal strains. *Saccharomyces boulardii* 27169, *Rhodotorula mucilaginosa* 27173, and *Zygosaccharomyces rouxii* LL12_088 were used as yeast strains. Fungal strains were sub-cultured on potato dextrose agar (PDA) (BD, Sparks, USA) plates, at 25 °C, at least two times, before being used for antimicrobial and membrane permeabilization testing.

#### 4.3.2. Minimal Inhibitory and Bactericidal Concentrations for Bacterial Strains

MICs for bacterial strains were determined using a broth microdilution protocol [88]. Two-fold dilutions of peptide solutions (0.256 ug/mL) were made in Mueller Hinton Broth (MHB) (Oxoid, Basingstoke, UK) in 96-well polypropylene clear round bottom microplates (Corning Inc., Corning, NY, USA). The 18-h bacterial cultures were diluted in MHB broth and inoculated in the microplates to obtain a final inoculum of 5 × 10^5^ colony-forming units (CFU)/mL. Final volumes in the wells were of 200 μL. Plates were then incubated for 16 h at 37 °C, except for plates containing *A. salmonicida* strains, which were incubated at 20 °C. MICs were defined as the lowest peptide concentrations that totally inhibited the visible growth of target bacteria in the wells [88]. Chloramphenicol (Sigma-Aldrich, St. Louis, MO, USA) was used as a positive control against *E. coli* ATCC 25922, and tests were considered valid if the MIC of this compound against this strain was 4 μg/mL [87]. After reading of the MICs, 20 μL of the wells that were showing no visible bacterial growth were inoculated on tryptic soy agar (TSA) (or BHI agar for *Listeria ivanovii*) plates, which were then incubated at the appropriate temperature for 24 h. MBCs were defined as the lowest peptide concentrations that inhibited at least 99.9% of bacterial growth, based on CFU counts relatively to the 5 × 10^5^ CFU/mL initial inocula in the wells [89]. Three independent repetitions were completed for each test, and two technical replicates were done for each independent repetition.

#### 4.3.3. Minimal Inhibitory and Fungicidal Concentrations for Fungal Strains

MICs were also tested using microdilutions in 96-well plates for fungal strains. The protocol used was roughly the same as that described for bacterial strains. RPMI 1640 (Sigma-Aldrich, St. Louis, MO, USA), buffered with MOPS (Fisher Scientific, Fair Lawn, NJ, USA) (0.164 M, pH = 7), was used instead of MHB, according to CLSI guidelines [90,91]. Potato dextrose broth was used for microplates inoculated with *Z. rouxii* yeast. For filamentous fungi, conidia were harvested from 7-day cultures and diluted in buffered RPMI 1640 to obtain a suspension containing 5 × 10^4^ CFU/mL, as assessed with a COUNTESS II FL automatic cell counter (Thermo Fisher Scientific, Waltham, MA, USA). Conidial suspensions were then inoculated in the microplates to reach a final inoculum of 2.5 × 10^4^ CFU/mL. For yeasts, vegetative cells were harvested from 2-day cultures and diluted in buffered RPMI 1640 to obtain a 5 × 10^3^ CFU/mL suspension, which was inoculated in plates to obtain a final inoculum of 2.5 × 10^3^ CFU/mL. Plates were then incubated for 24 (yeasts) or 48 (filamentous fungi) h at 25 °C. MICs were then defined as the lowest peptide concentrations that totally inhibited visible fungal growth in the wells. Natamycin (Sigma-Aldrich, St. Louis, MO, USA) was used as positive control for all fungal strains. MFCs were defined as the lowest peptide concentrations that inhibited visual growth of filamentous fungi or 99.9% of yeast growth (based on CFU counts), after having inoculated PDA plates with 20 μL from the wells, showing no fungal growth. Three independent repetitions were completed for each test, and two technical replicates were done for each independent repetition.

### 4.4. CD Measurements

CD measurements were performed with a JASCO J-815 spectrometer (Aviv Instruments, Lakewood, CA, USA). Peptides were dissolved in 10 mM PB (pH = 7.2), which solution was then used for spectrum acquisition or mixed with TFE) to obtain 25%, 50%, and 75% (*v*/*v*) TFE/PB solutions. Peptide concentration was 0.5 mg/mL for all solutions. Spectra were recorded at 25 °C in the 195–260 nm wavelength range (0.1 nm intervals), with a 0.2 mm path length cuvette. The 10 accumulations were averaged for each measurement. Spectra were smoothed with the J720/98 system program (Version 120C). CD spectra are presented as molar ellipticity θ (deg cm^2^ dmol^−1^) values plotted against wavelengths (nm). Spectra analysis was performed with the CONTIN algorithm included in the CDPro analysis Spectra Manager software (version 2.05.02, JASCO Corporation, Easton, MD, USA), using the SP43 reference.

### 4.5. Secondary Structure Predictions

PSIPRED 4.0 was used to predict the secondary structure of analogs 1 (SJGAP), 5, 6, and 7. Sequences of analogs 2, 3, and 4 were too short to be used as inputs for PSIPRED 4.0, as a minimum of 20 residues is required. Analog 8 structure was not specifically predicted with this tool, as it offers no option to include a C-terminal amidation. PSIPRED 4.0 is accessible from http://bioinf.cs.ucl.ac.uk/psipred/ (accessed on 30 January 2022).

### 4.6. Membrane Permeabilization Assays

#### 4.6.1. SYTO 9 and PI Assays

Peptides’ ability to permeabilize cytoplasmic membranes was assayed using the LIVE/DEAD^TM^ BacLight^TM^ Bacterial Viability kit (Invitrogen^TM^ L7012) (Thermo Fisher Scientific, Waltham, MA, USA). The 16-h cultures of *E. coli* ATCC 11229 and *M. luteus* LMA-272 and 24-h cultures of *R. mucilaginosa* 27173 were centrifuged (15 min, 10,000× *g*), and supernatants were disposed. Cells were then resuspended with sterile saline (0.85% NaCl) solution and adjusted to 2 × 10^8^ CFU/mL, based on the previously established correlations between optical density (600 nm) values and plate counts. Peptides diluted in sterile saline solution (50 μL) were mixed with microbial cell suspensions (50 μL) in Falcon^®^ polystyrene clear flat bottom 96 wells microplates (Corning Inc., Corning, NY, USA), so that final inocula were of 1 × 10^8^ CFU/mL. MICs and MBCs or MFCs (or 2x MICs when peptides had no bactericidal or fungicidal activity) were tested for peptide analogs 1, 5, 6, 7, and 8. If one of these peptides did not show activity against the tested microorganism, the concentrations that were selected were the same as those of the analogs active against that microorganism. Ethanol (50% *v*/*v*, final concentration in the wells) and sterile saline solution were, respectively, used as positive and negative controls. Microplates were then incubated at room temperature for 2 h, and 100 μL of SYTO 9/PI mixture, diluted in ultrapure Milli-Q™ water, according to the manufacturer’s guidelines, was then added to the microplate wells. Microplates were then incubated in the dark at room temperature for 15 min. After the incubation period, a BioTek Synergy H1 spectrophotometer (Agilent, Santa Clara, CA, USA) was used to read fluorescence values in each well (sequential scanning), with an excitation wavelength of 485 nm for both SYTO 9 and PI and emission wavelengths of 530 nm for SYTO 9 and 630 nm for PI. Blank wells consisting of peptide solutions, positive and negative controls only (without microbial cells), were included in each microplate, in order to subtract any absorbance due to peptides, ethanol, and saline solution. Three independent repetitions, including three technical replicates, were conducted for each microorganism and peptide concentration. Results are expressed as SYTO 9/PI absorbance ratios.

#### 4.6.2. The 260 nm Absorbing Material (DNA/RNA) Leak Assays

The determination of 260 nm absorbing material leakage from microbial cells induced by the peptides was based on a method described by Carson et al. [92] and Yasir et al. [93]. Microbial cell suspensions were prepared as described above, in Section 4.6.1. A total of 200 μL of these microbial suspensions were added to PCR grade DNA LoBind^®^ 1.5 mL tubes (Eppendorf, Hamburg, Germany), containing 200 μL of peptides, diluted in sterile saline solution. Final inocula were of 1 × 10^8^ CFU/mL and MICs and MBCs or MFCs (or 2x MICs when peptides had no bactericidal or fungicidal activity) were tested for peptide analogs 1, 5, 6, 7, and 8. 50 μL of these mixtures were withdrawn immediately after the addition of the cell suspensions to the peptide solutions (T0), diluted in saline solution (1:10), and filtered through 0.22 μM cellulose acetate filters. A total of 200 μL of these filtrates were then added in UV-STAR^®^ UV-transparent microplates (Greiner Bio-One, Frickenhausen, Germany). The same procedure was done after an incubation time of 120 min at room temperature. The 260 nm absorbance values were measured with a BioTek Synergy H1 spectrophotometer (Agilent, Santa Clara, CA, USA). Three independent repetitions, including three technical replicates, were conducted for each microorganism and peptide concentration.

### 4.7. Statistical Analysis

All statistical work was done with the SAS^®^ Studio software (version 3.8, SAS Institute Inc., Cary, NC, USA). One way ANOVA and Tukey’s HSD test were used to identify significant differences between treatments. Results were considered statistically significant when *p* < 0.05.

## 5. Conclusions

In conclusion, our experiments with SJGAP and its synthetic analogs revealed that their antimicrobial action was intimately related to their cationicity and hydrophobicity, and, although they induce membrane permeabilization in all three microorganisms tested, their mechanism of action appears to be at least partially related to intracellular targets. In addition, it is interesting to note that the MIC values determined for the most active SJGAP analogs (analogs 5 and 6, in particular) are equivalent or even slightly higher than those of magainin-2 (MICs of 50 ug/mL against *E. coli* ATCC 25922 and *P. aeruginosa* ATCC 27853) [94], which is an AMP that has been studied for almost three decades and is now commercially available. This demonstrates the potential application of these SJGAP analogs, as their significant antimicrobial activity has been established against strains of therapeutic and food interest. Further work, including molecular dynamics (MD) simulations and Fourier transform infrared spectroscopy (FTIR) experiments using different membrane models, will allow us to characterize more precisely the interactions between these peptides and microbial membranes, thus allowing to continue to investigate the mode of action of SJGAP and analogs thereof. This FTIR and MD work will allow us to more precisely investigate the involvement of the different residues constituting these peptides in the interactions with the membranes, as well as the effect of these interactions on the integrity of the latter. MD simulations could also be used to identify some intramolecular interactions between the residues of the analogs contributing to the stability of their secondary structure. Thus, more precise hypotheses on the mechanisms of antimicrobial action of these peptides at the molecular level could be put forward, allowing a better determination of their potential for application in a pharmaceutical or food industry context.

## Figures and Tables

**Figure 1 antibiotics-11-00297-f001:**
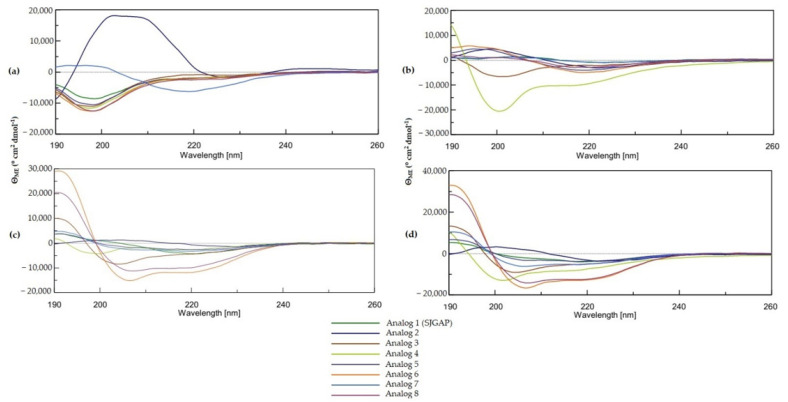
Circular dichroism (CD) spectra of peptide analogs 1–8 in (**a**) 10 mM phosphate buffer (PB), (**b**) trifluoroethanol (TFE) 25%, (**c**) TFE 50% and (**d**) TFE 75%.

**Figure 2 antibiotics-11-00297-f002:**
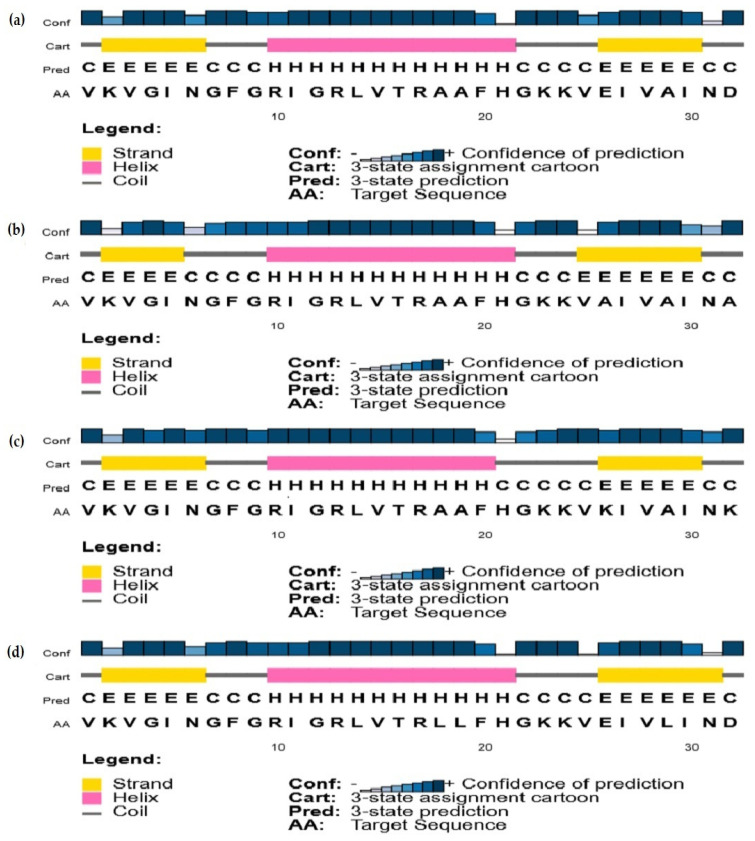
Secondary structures of (**a**) analog 1(SJGAP), (**b**) analog 5, (**c**) analog 6, and (**d**) analog 7, as predicted by PSIPRED.

**Figure 3 antibiotics-11-00297-f003:**
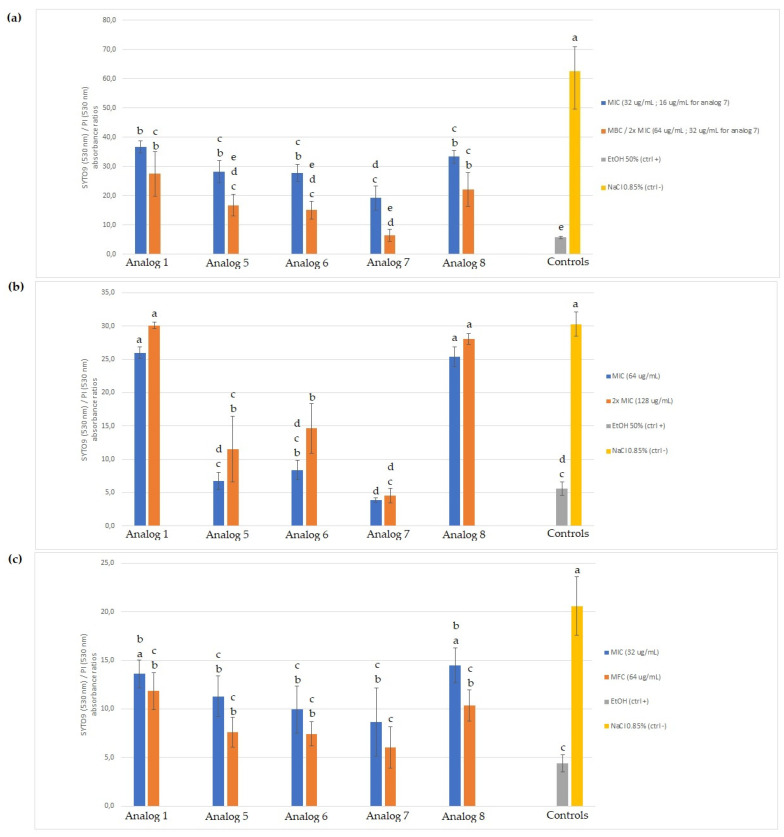
SYTO 9 (530 nm) / propidium iodide (PI) (630 nm) absorbance ratios, as obtained with the LIVE/DEAD^®^ BacLight Bacterial Viability kit for (**a**) *E. coli* ATCC 11229, (**b**) *M. luteus* LMA-272, and (**c**) *R. mucilaginosa* 27173. Each value is the average of three independent repetitions, performed in technical triplicates, and error bars represent standard deviations. Values sharing a common letter are not significantly different, according to Tukey HSD test (*p* ≤ 0.05).

**Table 1 antibiotics-11-00297-t001:** Net charges, isoelectric points, molar weights, and GRAVY indexes of the peptide analogs used in this study.

Peptide Analogs	Sequences	Net Charges	Isoelectric Points	Molar Weights (g/mol)	GRAVY Indexes
1 (SJGAP)	VKVGINGFGRIGRLVTRAAFHGKKVEIVAIND	+4	11.4	3436.07	0.272
2	VKVGINGFGRIG	+2	11.4	1216.45	0.558
3	IGRLVTRAAFHG	+2	12.1	1297.53	0.433
4	HGKKVEIVAIND	0	7.6	1322.53	−0.225
	VKVGINGFGRIGRLVTRAAFHGKKV**A**IVAIN**A**	+6	12.4	3334.02	0.603
6 *	VKVGINGFGRIGRLVTRAAFHGKKV**K**IVAIN**K**	+8	12.5	3448.21	0.247
7 *	VKVGINGFGRIGRLVTR**LL**FHGKKVEIV**L**IND	+4	11.4	3562.21	0.459
8 *	VKVGINGFGRIGRLVTRAAFHGKKVEIVAIN**D-NH_2_**	+5	11.9	3435.08	0.272

* Substituted or modified residues are highlighted in bold.

**Table 2 antibiotics-11-00297-t002:** Minimal inhibitory concentrations (MICs) and bactericidal concentrations (MBCs) of peptide analogs against seven bacterial strains.

Peptide Analogs	Antibacterial Activity (MIC; MBC) (μg/mL)						
	*E. coli* ATCC 25922	*E. coli* ATCC 11229	*P. aeruginosa* ATCC 27853	*A. salmonicida* 69 R3	*A. salmonicida* 69 R5	*M. luteus* LMA-272	*L. ivanovii* ATCC 19119
1 (SJGAP)	n.a. ^1^						
2	n.a.						
3	n.a.						
4	n.a.						
5	32; 128	32; 64	16; n.b.a. ^2^	32; 64	32; 128	64; n.b.a.	32; n.b.a.
6	64; 64	16; 32	16; 128	64; 128	64; n.b.a.	64; n.b.a.	32; n.b.a.
7	n.a.						
8	64; 64	32; 32	64; n.b.a.	128; n.b.a.	n.a.	n.a.	n.a.

^1^ No activity; ^2^ No bactericidal activity.

**Table 3 antibiotics-11-00297-t003:** MICs and minimal fungicidal concentrations (MFCs) of peptide analogs against six fungal strains.

Peptide Analogs	Antifungal Activity (MIC; MFC) (μg/mL)					
	*A. niger* 3071-13	*M. racemosus* LMA-722	*Paecilomyces* sp. 5332-9a	*R. mucilaginosa* 27173	*Z. rouxii* LL12_088	*S. boulardii* 27169
1 (SJGAP)	n.a. ^1^	n.a.	128; -	n.a.	128; n.f.a. ^2^	n.a.
2	n.a.					
3	n.a.					
4	n.a.	n.a.	n.a.	n.a.	64; n.f.a.	n.a.
5	n.a.	n.a.	32; n.f.a.	32; 64	64; n.f.a.	n.a.
6	n.a.	n.a.	32; n.f.a.	32; 64	64; n.f.a.	n.a.
7	n.a.	n.a.	16; n.f.a.	32; 64	64; n.f.a.	n.a.
8	n.a.	n.a.	64; n.f.a.	n.a.	64; n.f.a.	n.a.

^1^ No activity; ^2^ No fungicidal activity.

**Table 4 antibiotics-11-00297-t004:** Secondary structure contents of peptide analogs from the CONTIN algorithm analysis of CD spectra.

Peptide Analogs	Secondary Structure Contents (%)											
	PB (10 mM)			TFE 25%			TFE 50%			TFE 75%		
	α-Helix	β-Strands	Unrd *	α-Helix	β-Strands	Unrd *	α-Helix	β-Strands	Unrd *	α-Helix	β-Strands	Unrd *
1 (SJGAP)	6	35	37	4	42	33	6	40	32	7	38	33
2	6	44	35	5	40	33	4	41	34	5	38	35
3	5	33	39	6	37	35	15	33	30	18	29	31
4	6	32	39	5	32	37	4	39	35	6	34	33
5	6	33	39	5	41	32	6	41	32	8	38	32
6	6	30	41	6	42	31	39	17	24	41	14	28
7	8	33	36	4	42	33	7	38	33	16	30	33
8	7	30	40	5	40	34	25	23	31	36	18	26

* Unordered (random coils).

**Table 5 antibiotics-11-00297-t005:** Sequences and research interest of peptide analogs synthesized for this study.

Peptide Analogs	Sequence	Research Interest/Analog Modification
1 (SJGAP)	VKVGINGFGRIGRLVTRAAFHGKKVEIVAIND	Native SJGAP; model for this study
2	VKVGINGFGRIG	N-terminal segment of SJGAP
3	IGRLVTRAAFHG	Middle segment of SJGAP
4	HGKKVEIVAIND	C-terminal segment of SJGAP
5 *	VKVGINGFGRIGRLVTRAAFHGKKV**A**IVAIN**A**	Substitution of anionic residues with neutral alanine residues
6 *	VKVGINGFGRIGRLVTRAAFHGKKV**K**IVAIN**K**	Substitution of anionic residues with cationic lysine residues (+4 net charge)
7 *	VKVGINGFGRIGRLVTR**LL**FHGKKVEIV**L**IND	Substitution of alanine residues by more hydrophobic leucine residues
8 *	VKVGINGFGRIGRLVTRAAFHGKKVEIVAIN**D-NH_2_**	C-terminal amidation

* Substituted or modified residues are highlighted in bold.

**Table 6 antibiotics-11-00297-t006:** Identification and description of the bacterial and fungal strains used in this study.

Genus	Species	Strain	Type	Study Relevance
*Escherichia*	*coli*	ATCC 25922	Gram-negative	Human pathogen; reference strain [87]
*Escherichia*	*coli*	ATCC 11229	Gram-negative	Non-pathogenic *E. coli* strain
*Pseudomonas*	*aeruginosa*	ATCC 27853	Gram-negative	Human pathogen
*Aeromonas*	*salmonicida*	69 R3 ^a^	Gram-negative	Fish pathogen
*Aeromonas*	*salmonicida*	69 R5 ^a^	Gram-negative	Fish pathogen
*Micrococcus*	*luteus*	LMA-272 ^b^	Gram-positive	Human skin flora; opportunistic pathogen
*Listeria*	*ivanovii*	ATCC 19119	Gram-positive	Human pathogen
*Rhodotorula*	*mucilaginosa*	27 173 ^c^	Yeast	Human pathogen; food spoilage
*Saccharomyces*	*boulardii*	27 169 ^c^	Yeast	Food spoilage
*Zygosaccharomyces*	*rouxii*	LL12_088 ^d^	Yeast	Food spoilage
*Aspergillus*	*niger*	3071-13 ^e^	Filamentous fungi	Food spoilage
*Mucor*	*racemosus*	LMA-722 ^b^	Filamentous fungi	Food spoilage; opportunistic pathogen
*Paecilomyces*	sp.	5332-9 ^e^	Filamentous fungi	Food spoilage; opportunistic pathogen

^a^ Strain provided by Professor Steve Charette (Labo Charette, Université Laval, QC, Canada); ^b^ strain collection of Professor Steve Labrie (Laboratoire de Microbiologie Alimentaire, Université Laval, QC, Canada); ^c^ strain collection of General Mills Yoplait (Boulogne-Billancourt, France); ^d^ strain collection of Professor Christian Landry (Université Laval, QC, Canada); ^e^ strain collection of Professor Denis Roy (Université Laval, QC, Canada).

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
