# Peer review of "Determination of the Relationships between the Chemical Structure and Antimicrobial Activity of a GAPDH-Related Fish Antimicrobial Peptide and Analogs Thereof"

_antibiotics, 2022, doi:10.3390/antibiotics11030297_

Round 1

Reviewer 1 Report

antibiotics-1600739-peer-review

Determination of the Relationships Between the Chemical

Structure and Antimicrobial Activity of a GAPDH-Related Fish

Antimicrobial Peptide, and Analogs Thereof

Some suggestions are given below

Section 2.2. Antimicrobial Activity

  • In Tables 2 and 3, please include the MICs and MBC/MFC values of the reference standards or antibiotics.

  • Please change the hyphen (-) in tables 2 and 3 to not determined (nd) or whatever corresponds. Please be more specific.

  • Please include some of the following bibliographical references, which give parameters or ranges of antimicrobial activity.

Consider the bibliographical references suggested briefly in the discussion.

Plese see reference Kuete and Efferth 2010, where antibacterial activity parameters are given for extracts and pure compounds. Extract: significant (MIC<100μg/ml), moderate (100<CMI≤625μg/ml) or weak (CMI>625 μg/ml).

For compounds, this stringent endpoints criteria were: significant (MIC<10μg/ml), moderate (10<MIC≤100 μg/ml) and low or negligible (MIC > 100 μg/ml) Kuete V, Efferth T (2010) Cameroonian medicinal plants: pharmacology and derived natural products. Front Pharmacol 1:123"

Please see Journal of Ethnopharmacology Volume 100, Issues 1–2, 22 August 2005, Pages 80-84. Perspective paper Medicinal plants and antimicrobial activity J.L.RíosM.C.Reciohttps://doi.org/10.1016/j.jep.2005.04.025

“whereas the presence of activity is very interesting in the case of concentrations below 100 µg/ml for extracts and 10µg/ml for isolated compounds”.

After the suggested changes the manuscript should be accepted for publication                 

Author Response

Reviewer # 1:

Some suggestions are given below

Section 2.2. Antimicrobial Activity

In Tables 2 and 3, please include the MICs and MBC/MFC values of the reference standards or antibiotics.

In each of our independent MICs/MBCs assay repetitions, we tested chloramphenicol (reference antibiotic) against the reference strain E. coli ATCC 25922 to confirm the validity of the results obtained (see section 4.3.2.), as recommended in the official procedure for this type of antimicrobial testing (Wiegand, I., K. Hilpert, and R.E. Hancock, Agar and broth dilution methods to determine the minimal inhibitory concentration (MIC) of antimicrobial substances. Nat Protoc, 2008. 3(2): p. 163-75 DOI: 10.1038/nprot.2007.521). The antimicrobial tests were considered valid if chloramphenicol showed a MIC of 4 μg/mL for this strain.

If you wish, we can include this 4 μg/mL value in table 2, but perhaps we could include this value in section 2.2 instead to make it clearer (please see lines 159-160), which would avoid adding a line containing a single piece of data in table 2.

We also added precision in table 6 to specify that E. coli ATCC 25922 was a reference strain.

Please change the hyphen (-) in tables 2 and 3 to not determined (nd) or whatever corresponds. Please be more specific.

Please see tables 2 and 3 and lines 163 and 166.

Please include some of the following bibliographical references, which give parameters or ranges of antimicrobial activity.

Consider the bibliographical references suggested briefly in the discussion.

Please see reference Kuete and Efferth 2010, where antibacterial activity parameters are given for extracts and pure compounds. Extract: significant (MIC<100μg/ml), moderate (100<CMI≤625μg/ml) or weak (CMI>625 μg/ml).

For compounds, this stringent endpoints criteria were: significant (MIC<10μg/ml), moderate (10<MIC≤100 μg/ml) and low or negligible (MIC > 100 μg/ml) Kuete V, Efferth T (2010) Cameroonian medicinal plants: pharmacology and derived natural products. Front Pharmacol 1:123"

Please see Journal of Ethnopharmacology Volume 100, Issues 1–2, 22 August 2005, Pages 80-84. Perspective paper Medicinal plants and antimicrobial activity J.L.RíosM.C.Reciohttps://doi.org/10.1016/j.jep.2005.04.025

“whereas the presence of activity is very interesting in the case of concentrations below 100 µg/ml for extracts and 10µg/ml for isolated compounds”.

After the suggested changes the manuscript should be accepted for publication

Please see lines 356-361.

Reviewer 2 Report

This manuscript focuses on the structure-activity relationship of GAPDH-related peptides and their analogs. The activity is closely related to hydrophobicity and net charge. As a new family of antibiotics that aims for multi-drug resistant microorganisms, antimicrobial peptides showed great potential. The reveal of structure and physicochemical properties will help to design new drugs with better bioactivities. The overall quality of this paper is acceptable, though there are still several questions that need to be addressed:

  1. From the experimental design, SJGAP with 32 residues was split into 3 analogs (Analog 2, 3, and 4) evenly, each containing 12 amino acid residues, rather than selecting the point at the loop between strand and helix motifs. When we look at the predicted secondary structure, this type of truncated subsets made the helix motif not intact (e.g. analog 3 started with IGRL so the beginning of the helix was missing). Have authors considered this situation and can testify that this design is the optimal selection for truncation?
  2. Have the authors considered the intra-relationship among the residues which can stabilize the peptides? And which residues can stretch towards the outside to react with the cell surface? If the key residues are not exposed to the reaction sites, then the analysis of hydrophobicity and charges will not give strong support to the conclusion. The reviewer suggested NMR or crystallography may be applied to get at least one of the actual structures instead of simulation of all.
  3. Line 128, Analog 7 is not the second highest of GRAVY value. Less than analog 2 and 5.
  4. In the Conclusions session, the future work about molecular dynamics simulations and IR (FT-IR or NIR or RAMAN) was not mentioned anywhere else in the paper. It would be better to describe it a little bit in the introduction or specify how these prospective studies will benefit this whole project.

Author Response

Reviewer # 2:

This manuscript focuses on the structure-activity relationship of GAPDH-related peptides and their analogs. The activity is closely related to hydrophobicity and net charge. As a new family of antibiotics that aims for multi-drug resistant microorganisms, antimicrobial peptides showed great potential. The reveal of structure and physicochemical properties will help to design new drugs with better bioactivities. The overall quality of this paper is acceptable, though there are still several questions that need to be addressed:

  1. From the experimental design, SJGAP with 32 residues was split into 3 analogs (Analog 2, 3, and 4) evenly, each containing 12 amino acid residues, rather than selecting the point at the loop between strand and helix motifs. When we look at the predicted secondary structure, this type of truncated subsets made the helix motif not intact (e.g. analog 3 started with IGRL so the beginning of the helix was missing). Have authors considered this situation and can testify that this design is the optimal selection for truncation?

This comment is interesting, and we thought about the different possibilities of analogs before deciding on these. We thought it was very important to have 3 segments of identical length so that the comparisons between these 3 segments (N-terminal, central and C-terminal) would be as meaningful as possible. Since the reference peptide (SJGAP, analog 1) is composed of 32 amino acid residues, we thought that choosing 12-residue segments would provide 3 segments of identical length that overlapped by only 2 amino acids. Moreover, the resulting analog 3 almost perfectly represents the central alpha helix of analog 1, except for the R10 residue at the very beginning of the helix, as you point out. However, since the different secondary structure motives within a peptide or a protein can be influenced by the residues that precede and follow them, which means that the alpha helix may not be formed by exactly the same residues in the native SJGAP as in a truncated segment, we believe that this difference of only one residue in the alpha helix represents an acceptable compromise in order to have 3 segments of equal length.

Finally, we also thought that this way of segmenting SJGAP was interesting, because the C-terminal segment (analog 4) is highly homologous to the antimicrobial peptide named AMGAP that was identified in a previous publication and mentioned in this manuscript.

  1. Have the authors considered the intra-relationship among the residues which can stabilize the peptides? And which residues can stretch towards the outside to react with the cell surface? If the key residues are not exposed to the reaction sites, then the analysis of hydrophobicity and charges will not give strong support to the conclusion. The reviewer suggested NMR or crystallography may be applied to get at least one of the actual structures instead of simulation of all.

Thank you for highlighting this point. We are aware that these considerations are very important and several experimental and bioinformatics investigations done on the same peptide analogs will be the subject of a subsequent publication in order to better understand the role of the different residues in their interactions with the membranes and the effect of these interactions on their integrity. We have clarified the conclusion (please see lines 720-727) to address these points more explicitly.

Moreover, the circular dichroism experiments that we conducted in conjunction with the secondary structure predictions provide a very good approximation of the structure of the analogs and obtaining the exact structure of a single one of these analogs by NMR or crystallization, while clearly interesting, is not required, in our opinion, to support the demonstrated relationships between charge and hydrophobicity of analogs and their antimicrobial activity. Indeed, we have not yet put forward a hypothesis concerning the precise role of the different residues, but rather we have reflected, on the basis of relevant literature, on the physico-chemical properties of the peptides in order to better understand their initial interactions with microbial membranes.

  1. Line 128, Analog 7 is not the second highest of GRAVY value. Less than analog 2 and 5.

Please see lines 128-129.

  1. In the Conclusions session, the future work about molecular dynamics simulations and IR (FT-IR or NIR or RAMAN) was not mentioned anywhere else in the paper. It would be better to describe it a little bit in the introduction or specify how these prospective studies will benefit this whole project.

Absolutely. Please see lines 716-727.